# Effects of Initial Surface Evaporation on the Performance of Fly Ash-Based Geopolymer Paste at Elevated Temperatures

**Thathsarani Kannangara \*, Maurice Guerrieri, Sam Fragomeni and Paul Joseph** 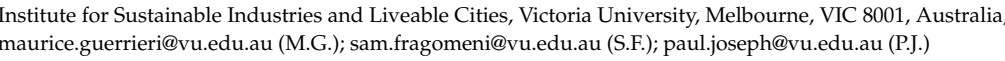

Institute for Sustainable Industries and Liveable Cities, Victoria University, Melbourne, VIC 8001, Australia;
maurice.guerrieri@vu.edu.au (M.G.); sam.fragomeni@vu.edu.au (S.F.); paul.joseph@vu.edu.au (P.J.)
\* Correspondence: a.kannangara@live.vu.edu.au

**Abstract:** Geopolymer concrete is a valuable and alternative type of concrete that is free of traditional cement. Generally, geopolymer concretes require a source material, which is rich in silicon and aluminum. Furthermore, fly ash-based geopolymer concretes have been proven to have superior fire resistance, primarily due to their ceramic properties, and are inherently environmentally-friendly given their zero-cement content. This paper presents the effects on initial evaporation on the performance of fly ash-based geopolymer pastes after exposure to elevated temperatures of 400 °C and 800 °C. The fly ash (FA) samples used in the present study included: Gladstone and Gladstone/Callide. The results for sealed samples placed in the oven during curing were much more consistent than the samples that were not kept covered. In addition, Gladstone fly ash-based geopolymer samples that were sealed recorded an initial maximum compressive strength reading of ca. 75 MPa, while sealed Gladstone/Callide fly ash-based geopolymer samples, of the same mix design, only recorded an initial maximum compressive strength reading of ca. 50 MPa (both subjected to oven curing at 60 °C for 24 h). However, Gladstone/Callide fly ash-based geopolymer samples exhibited a significant strength gain, ca. 90 MPa, even after being subjected to 400 °C.

**Keywords:** geopolymer; fly ash content; surface evaporation; residual strength

## 1. Introduction

Geopolymer (GP)-based concrete, first brought into light in the 1970s by a French scientist Joseph Davidovits, is an environmentally-benign material with a relatively lower carbon footprint compared to conventional concrete made from ordinary Portland cement (OPC) [1–4]. Referred to as a next generation concrete, and also chemically classified as a polysiliate, GP can be considered as a versatile, inorganic paste similar to zeolite materials with an amorphous microstructure [5–7]. Geopolymers are generally formed through a reaction taking place between alumino-silicate minerals and alkaline solution at ambient temperatures. During this 'geopolymerization' reaction, –Si–O–Al–O– bonds, similar to amorphous feldspar, are formed when the source material reacts with the alkaline activator [6,8–11]. The alkaline activator, which is another important factor that determines the performance of GP-based materials, can be sodium hydroxide (NaOH) or potassium hydroxide (KOH), which is taken in conjunction with other compounds such as sodium silicate ($Na_2SiO_3$) or potassium silicate ($K_2SiO_3$), sodium sulfate ($Na_2SO_4$), etc. [12,13]; however, due to the availability and cost effectiveness, NaOH and $Na_2SiO_3$ solutions are generally employed [7,14–18].

There are a number of reports in the literature [4,7,19–22] highlighting that the binder material should be one that is rich in aluminum and silicon. Other related studies [18,23–26] have indicated that components such as FA, slag, and metakaolin are often used as the source material(s) as they are rich in oxides of silica and aluminum. It has been also reported that metakaolin is the most commonly used source material due to its consistent composition; however, it is relatively expensive [27]. Finely powdered FA, on the other

hand, has since become a more popular source material. In addition to the fact that it is a waste residue from coal-based power-plants (thus making it more sustainable), FA-based GP concretes exhibit increased residual strengths after exposure to elevated temperature conditions (even up to 800 °C) [28–30]. It was also reported that the metakaolin-based pastes underwent a significant decrease (34%), while FA-based GP pastes displayed a 6% increase in the residual strength after exposure to 800 °C [31]. The gel microstructure of FA-based GP concretes generally control the internal moisture content, and thus plays a major role in increasing the resistance to spalling when exposed to elevated temperatures [32,33].

A study to evaluate the strength of FA-based geopolymer pastes in comparison to OPC pastes, after being exposed to elevated temperature levels, revealed that the former was more resistant to degradation than the latter [34]. In addition, it was also established that the residual compressive strength remained somewhat constant at temperatures 800–1000 °C for GP-based concretes, while the compressive strength fell to zero at 800 °C for the OPC-based concrete [35]. In another report, the compressive strength of FA-based GP pastes, reinforced with carbon and basalt fibers at 0.5, 1, and 1.5% by weight, after being exposed to temperatures of 200, 400, 600, and 800 °C was tested [26]. The results revealed better compressive strengths, lower volumetric shrinkage, and lower mass losses from samples comprising of 1% fibers. Furthermore, specimens reinforced with carbon fibers performed better compared to specimens incorporating basalt fibers over the temperature range that was employed. Bazan et al. [36] studied the influence of melamine and steel fibers on the compressive and bending strengths of FA-based GP mortar specimens. It was found that reinforcing the specimens with either of the fibers improved both types of strengths, with steel fibers having a better ability to dissipate stress during three-point bending tests, and samples with melamine fibers were shown to increase resistance to axial compression. In addition, the use of optimal curing regimes resulted in further improvements in the mechanical properties [22,37]. Aldred and Day [20] reported that the temperature pertaining to the curing regime is critical for the development of strength in GP-based concretes, with acceptable levels of initial strength were obtained when samples were subjected to higher temperatures above room temperature. Other studies have also shown that curing test samples at higher temperatures of 60–70 °C increased the compressive strength along with lower permeability level attributes [9,38]. Experiments were also carried out to gauge the effects of temperature on the strengths of the test specimens; for instance, at ambient and 60 °C for 24 h [39].

Through the current study, it was identified that generally hot curing conditions were most suited, with samples producing higher strength readings compared to samples subjected to ambient-temperature curing. Unlike ordinary cement-based concretes, GP-based concretes generally require high-temperature curing to achieve the bonding between constituent molecules, and hence result in the formation of a stronger microstructures [40]. However, it was also found that curing at very high temperatures of (i.e., typically over 90 °C) had an adverse effect on the development of desirable physical properties. For instance, the compressive strengths of specimens cured above 90 °C were found to be noticeably reduced in comparison to specimens cured at temperatures below 90 °C. This was assumed to be due to the continuous loss of moisture during curing at elevated temperatures, which also led to shrinkage cracks due to excessive drying, thus, producing weaker specimens [41].

It was reported that ambient temperature curing of GP-based concrete tends to produce low strength specimens in the initial phases [42]; however, these specimens significantly gain strength with time. Furthermore, it was noted that by increasing the curing time to at least 20 h, the rate of the geopolymerization reaction could be enhanced, thus producing specimens with improved strengths [42]. The report also indicated that while curing GP-based materials at temperatures between 40–80 °C did achieve the optimum compressive strength readings, it was also necessary to cure specimens for longer periods to obtain enhanced mechanical properties such as compressive, tensile, and flexural strengths [16–18,43]. In addition, GP-based materials are reported to have achieved almost

their full compressive strength during the first 24–48 h after casting and heat curing. This can be attributed to the achievement of the near complete polymerization process, and any further curing can only result in diminished returns [40,41,44,45].

Apart from its initial strength, GP-based concretes are also known to behave exceptionally well in high temperature/fire scenarios. For instance, when test samples (FA-based GP concrete) were heated up to 750 °C, they revealed the good strength characteristics of the test specimens at elevated temperatures compared to counterparts made from conventional concrete [46]. In another study by Mane and Jadhav [47], the residual mass loss properties and residual strengths of FA-based GP concretes (with low calcium contents) and mortars were monitored after they were exposed up to 500 °C. It was shown that the test specimens displayed 84% more strength in comparison with OPC-based counter parts. It was also noticed that while the OPC-based mortars displayed only strength losses as the temperature was increased, the compressive strength of GP-based mortars increased upon reaching a temperature of 100 °C, after which it was seen to be diminished.

GP-based concretes are indeed considered as a better alternative to conventional concretes in terms of developing a sustainable construction industry, with a 61% reduction in global warming potential and a 9.4% improvement in the human health category [48]. Partial- or full-replacement of cement using cementitious materials such as FA has been found to be one of the most effective methods in reducing the carbon footprint of concrete [49,50]. While numerous studies on the behavior, based on durability, strength, and other such mechanical properties of GP concretes at various exposure levels are available, research on the effects of the initial surface evaporation on FA-based GP pastes is very limited. In addition, specific literature precedents on a systematic comparison of the residual strengths of FA-based GP pastes (both Gladstone and Gladstone/Callide) are also severely lacking.

The novelty of the current work stems from the fact that we have endeavored to address the above knowledge gaps in the subject area. For instance, in the present study, we formulated a total of ten novel mix designs of FA-based GP pastes, with the main intention of monitoring the influence of the extent of the initial surface evaporation on the performance of these materials. Here, we also carefully chose two different experimental approaches that primarily differed in the extent of surface evaporation during the curing process. This was achieved by placing the samples in polymeric bags (i.e., sealed), or kept exposed (i.e., unsealed) for the entire duration of the curing regime. The effect of moisture retention on the performance of the cured samples were mainly evaluated through several tests for measuring the density, setting times, initial compressive strengths, residual strengths, and mass losses after exposure to elevated temperatures. Furthermore, testing was conducted on both varieties of FA (Gladstone and Gladstone/Callide) as the chemical compositions of these two types have nominal variations. Therefore, the effects of the changes in the chemical composition of different source materials on the test parameters were also evaluated. Thus, the relevant data and additional knowledge, gathered through the present study, will provide guidelines for the safe design of FA-based geopolymer pastes, especially when they are exposed to high temperatures. To our knowledge, there are no previously published systematic studies pertaining to the influence of moisture evaporation on FA-based GP paste samples that are made from different mix compositions.

## 2. Materials and Methods

This section provides the experimental details such as materials, sample preparation, curing procedures, and testing of cubical samples made from GP-based pastes. Here, parameters such a density, setting times, compressive strengths before and after exposure to elevated temperatures, and mass losses for sealed and unsealed samples of Gladstone FA-based GP mixtures were recorded. In addition, all of the above-mentioned parameters were also measured for Gladstone/Callide FA-based GP mixtures. All tests were conducted under laboratory conditions using fresh and hardened specimens, and measurements were conducted in triplicate, and the average values are quoted. Statistical information

relating to the empirical parameters such as errors and standard deviations were also added where applicable.

## 2.1. Materials

The main constituents in the GP mixture were FA, sodium hydroxide, and sodium silicate. Furthermore, two different kinds of FA were used such as, Gladstone FA, and Gladstone/Callide FA. Gladstone FA was class F-low calcium FA, light grey in color, and with a particle size ranging from 1–8 μm whereas the other variety (Gladstone/Callide FA- class F-low calcium FA) was darker in color. Both FA materials were found to be similar in texture to Ordinary Portland Cement (OPC). Their chemical compositions are given in Table 1. The fineness percentages, passing the 45 μm sieve, were recorded to be approximately 86% and 80% for Gladstone FA and Gladstone/Callide FA, respectively. Sodium hydroxide and sodium silicate were used as components for the alkaline activator. Sodium silicate solution was type D, having a ratio of silica to sodium oxide of 2.0. The sodium hydroxide solution had a strength of 60 w/v% and a molarity of 8 (i.e., mols/dm$^3$).

**Table 1.** Chemical compositions of FA.

| Oxide | Oxide (wt. %) | |
| --- | --- | --- |
| | **Gladstone FA** | **Gladstone/Callide FA** |
| $SiO_2$ | 51.1 | 52.8 |
| $Al_2O_3$ | 25.6 | 28.8 |
| $Fe_2O_3$ | 12.5 | 9.99 |
| CaO | 4.30 | 2.70 |
| $K_2O$ | 0.70 | 0.45 |
| MgO | 1.45 | 1.13 |
| $Na_2O$ | 0.77 | 0.44 |
| $TiO_2$ | 1.32 | 1.71 |
| BaO | 0.09 | 0.08 |
| $SO_3$ | 0.24 | 0.17 |
| $P_2O_5$ | 0.89 | 0.49 |
| MnO | 0.15 | 0.08 |

## 2.2. Specification of the Mix Design

The details of the ten novel mix designs that were employed for this study are given in Table 2.

**Table 2.** Details regarding the mix designs of FA–based pastes *.

| Sample ID | Sample Composition for 1 kg of FA-(Alkaline Solution/FA Ratio; $Na_2SiO_3$/NaOH) | Sodium Silicate Grade D (kg) | 8 M NaOH (kg) | Total Weight (kg) |
| --- | --- | --- | --- | --- |
| GP 01 | GP-0.40; 0.50 | 0.133 | 0.267 | 1.40 |
| GP 02 | GP-0.40; 1.00 | 0.200 | 0.200 | 1.40 |
| GP 03 | GP-0.40; 1.75 | 0.255 | 0.145 | 1.40 |
| GP 04 | GP-0.40; 2.00 | 0.267 | 0133 | 1.40 |
| GP 05 | GP-0.40; 2.50 | 0.286 | 0.114 | 1.40 |
| GP 06 | GP-0.57; 0.50 | 0.190 | 0.380 | 1.57 |
| GP 07 | GP-0.57; 1.00 | 0.285 | 0.285 | 1.57 |
| GP 08 | GP-0.57; 1.75 | 0.363 | 0.207 | 1.57 |
| GP 09 | GP-0.57; 2.00 | 0.380 | 0.190 | 1.57 |
| GP 10 | GP-0.57; 2.50 | 0.407 | 0.163 | 1.57 |

* Here for both types of fly ash, the same proportions were used (i.e., 1 kg each).

### 2.3. Mixing Procedure

Sodium hydroxide and sodium silicate were first measured and mixed using a hand-held stirrer bar until a clear, transparent solution was observed (ca. 2 min). This mixture was kept aside for a few minutes before stirring with the fly ash component. This method was essentially adopted from a previously published work by Hardjito and Rangan [51]. The required amount FA was measured and mixed to the liquid solution using a Breville mixer (ca. at 50 rpm for 2 min), and then for a further 3 min at ca. 85 rpm, after which, 25 mm$^3$ specimens were cast.

### 2.4. Curing Regime

After casting, the cubes with dimensions of $25 \times 25 \times 25$ mm$^3$ were immediately subjected to curing. For this, a WEISS WVC Series Temperature and Climatic Test Chamber was employed. The cubes were kept at ca. 60 °C for 24 h. Essentially, the curing was conducted using two methods. Basically, these methods were designed to test the influence of the extent of the surface evaporation of water on the mechanical properties of the samples; for example, compressive and residual strengths. Unsealed samples were placed in the oven at 60 °C immediately after casting. Once hardened, samples were removed from the mold and placed back in the oven for a total of 24 h. Sealed samples, on the other hand, were placed in the polymeric bag, soon after casting, and were subjected to an elevated temperature by placing them in an oven maintained at 60 °C (Figures 1 and 2). Gladstone/Callide FA specimens were only subjected to the sealed-curing procedure. It is to be noted here that none of the samples were subjected to a rest period (i.e., after casting), and all samples were immediately subjected to heat curing.

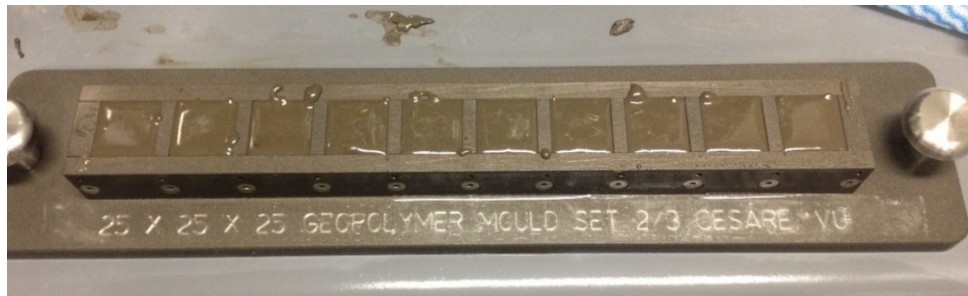

**Figure 1.** Gladstone FA GP casting (unsealed specimens).

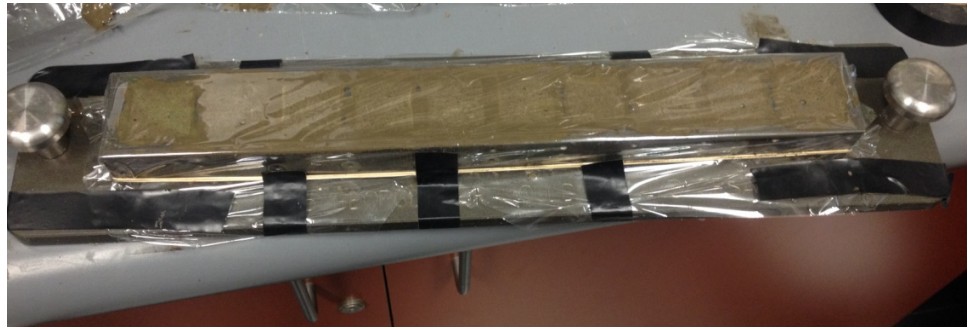

**Figure 2.** Gladstone FA GP casting (sealed specimens).

### 2.5. Testing of Fresh and Hardened Specimens

The density of Gladstone FA and Gladstone/Callide FA mixtures were determined, in conformance with AS 1012.5:2014 [52]. In addition, the setting times were also recorded using visual observations. For hardened specimens, the compressive and residual strengths were investigated and conducted in accordance with AS 1012.9.2014 [53].

The compressive strength of the test specimens was determined using a 100 kN Instron 1195 testing machine at a loading rate of $20 \pm 2$ MPa/min. The Gladstone FA and Gladstone/Callide FA cubes were tested after 24 h of casting and curing for initial compressive strength. The samples that were tested for residual strength were subjected to heating in a muffle furnace at a steady rate of 10 °C/min to achieve the target temperatures of 400 °C and 800 °C. Upon reaching these target temperatures, samples were held at the required temperature for a period of one hour with a view to establishing the thermal equilibrium (i.e., to achieve a constant temperature throughout the samples). The samples were then permitted to cool down to room temperature (ca. 25 °C) before they were tested for strength. Evidently, testing the specimens for residual compressive strengths after they had been cooled down to the ambient temperature denotes the lower bound strength values compared to the stressed residual tests, where samples were tested whilst subjected to elevated temperatures, for example, as reported elsewhere [54]. The corresponding mass losses were measured using an electronic balance, where cube samples were weighed before and after exposure to elevated temperatures, and from the readings decrements, if present, were recorded.

## 3. Results and Discussion

The Gladstone FA-based GP samples exhibited higher density readings compared to Gladstone/Callide FA-based specimens. For example, the former set of samples had density values ranging from 2396 kg/m$^3$ and 2154 kg/m$^3$, and for the latter set, it was between 2059 kg/m$^3$ and 1870 kg/m$^3$. It was also found that the most-dense Gladstone FA-based GP paste was GP 10, and the most-dense Gladstone/Callide FA-based GP paste was GP 05.

When considering the setting times, it was observed that Gladstone FA-based GP pastes remained in liquid state for about 30 min, whereas the other category of pastes set quicker than 30 min. The initial setting times for samples such as GP 01, 02, and 03 of the Gladstone/Callide FA-based GP pastes were between 2 and 4 min, and hence cubes of these samples could not be cast. Figure 3 shows a comparison of the morphology of Gladstone FA-based GP 01 and Gladstone/Callide FA-based GP 01 samples 5 min after casting. This could be attributed to the changes in the pH value of the solution with the addition of NaOH. A higher hydroxide content has the ability to increase the alkalinity of a solution, thus, increase the pH value of a solution, and vice versa. In the case of GP-based materials, it was previously reported that at lower values of pH, the GP mixture exhibited enhanced fluidity (i.e., more workable), while at relatively higher pH values, the mixtures exhibited increased viscosity, thus resulting in the accelerated setting [55]. This could explain why GP 01, 02, and 03, which had a decreasing ratio of $Na_2SiO_3$ to NaOH (i.e., a higher hydroxide content), thus having an increase in pH, set quicker than GP 04 and 05.

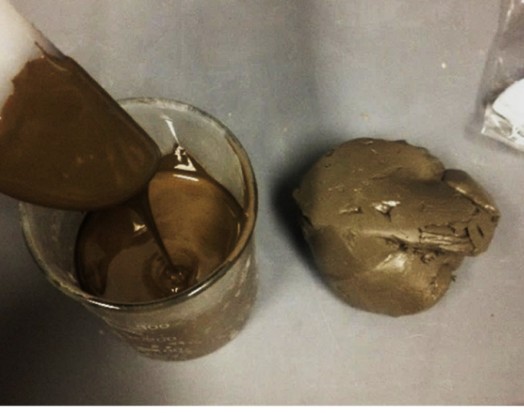

**Figure 3.** A picture indicating the morphological features of the mixtures: Gladstone FA GP01 (**left**)/Gladstone/Callide FA GP01 (**right**).

It was interesting to note that the above effect did not occur in the case of Gladstone FA-based matrices. They in fact exhibited longer setting times even with the variation in the ratios of $Na_2SiO_3$ to NaOH. This can be attributed to the difference in amounts of the oxides of aluminum and silicon in the formulations of the two FA materials. It was reported that aluminum can exert a significant influence on the setting times of the GP pastes [56]. For instance, it is established with lower $SiO_2:Al_2O_3$ ratios produced shortened setting times, and that slight variations in the Si and Al concentrations can result in substantially different setting times for the GP mixtures. As given in Table 1, Gladstone FA had a slightly higher ratio of approximately 2.0 as opposed to 1.8 for Gladstone/Callide FA, which indicates a longer setting time in the latter case. It was also reported [57] that a higher silica content in geopolymers necessitated a higher water content. In addition to this, presence of a higher content of aluminum within the matrix can initiate quick condensation and thus accelerate the geopolymerization reaction [58,59]. This supports the findings from the present study, as Gladstone/Callide FA does indeed have a higher silica and aluminum content compared to Gladstone FA.

In Table 3, a comparison of the initial compressive strengths among the sealed and unsealed Gladstone FA-based GP specimens are tabulated and are graphically presented in Figure 4. It can be seen that unsealed specimens produced strengths ranging from approximately 15 to 58 MPa, while sealed specimens recorded much higher strengths ranging from approximately 23 to 74 MPa, with the highest strength recorded from GP05 for both sealed and unsealed specimens.

**Table 3.** Average compressive strengths (MPa) at 24 h between unsealed and sealed Gladstone FA GP cubes.

| Sample ID | Unsealed Samples | | Sealed Samples | |
|---|---|---|---|---|
| | Strength | STDEV | Strength | STDEV |
| GP 01 | 14.83 | 7.24 | 28.76 | 4.67 |
| GP 02 | 14.88 | 6.75 | 41.31 | 3.28 |
| GP 03 | 24.00 | 7.91 | 67.06 | 3.31 |
| GP 04 | 57.40 | 8.98 | 67.95 | 1.34 |
| GP 05 | 57.97 | 7.85 | 74.48 | 3.41 |
| GP 06 | 20.80 | 4.35 | 22.67 | 3.64 |
| GP 07 | 40.27 | 3.78 | 41.98 | 2.57 |
| GP 08 | 42.13 | 4.17 | 54.77 | 2.12 |
| GP 09 | 48.37 | 2.19 | 55.38 | 1.32 |
| GP 10 | 55.40 | 1.71 | 58.13 | 1.72 |
| Minimum | 14.83 | 1.71 | 22.67 | 1.32 |
| Maximum | 57.97 | 8.98 | 74.48 | 4.67 |
| Average | 37.61 | - | 51.25 | - |

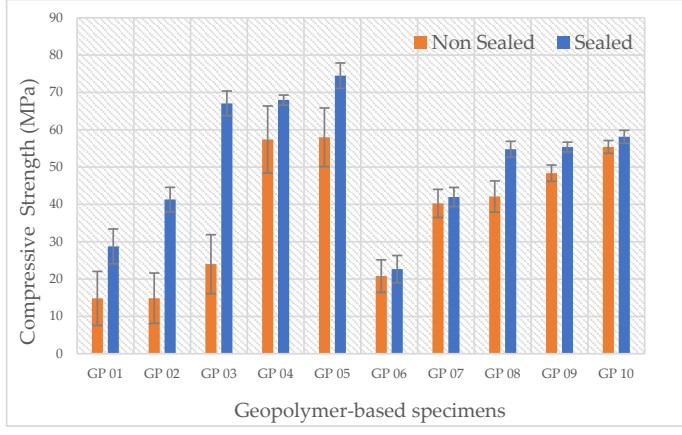

**Figure 4.** A plot of the initial compressive strengths among the unsealed and sealed specimens.

An average initial strength increase of approximately 25% was recorded in the sealed specimens compared to unsealed specimens, where sealed specimens exhibited initial strength readings of up to approximately 74 MPa. This constitutes a direct evidence that initial surface evaporation plays a crucial role in the development of strength of GP pastes. The difference in strength can be attributed to the extent of the initial dehydration of the matrices during the curing process. The absence of sufficient levels of fluidity for the dissolution and gelation processes will hinder any further developments in strength during the geopolymerization reaction. Consequently, this could lead to the breakdown of the granular structure of the matrix, as reported earlier [60,61]. It has also been reported that in most cases, GP achieves 70% of its strength during 12 h [62]; however, other studies [63] stated that this 70% can be reached within 3–4 h of curing. Therefore, barricading the samples from excess evaporation during this initial strength gain period will indeed produce more consistent results and relatively higher values.

The degradation of strength could also stem from the occurrence of a carbonation process. During its early setting stages, especially when the specimens are exposed to ambient conditions, $CO_2$ can have an easy access to, and also diffuse rapidly through, the matrix. This can lead to the production of sodium bicarbonate, which in turn reduces the pH value, thus creating a more acidic environment, resulting in decreased formation of the alumino-silicate gel. It has been reported previously [64,65] that for binders with relatively low calcium contents, higher alkaline concentrations are essential for the development of strength.

Moreover, it was noted that as the ratio of the silicate to hydroxide was increased from 0.5 to 2.5, both sealed and unsealed specimens recorded increased compressive strengths (Table 3). In the case of unsealed specimens, the initial compressive strengths increased from approximately 14 MPa to 58 MPa (for GP 01 to GP 05) and from 21 MPa to 55 MPa (for GP06 to GP 10), and for sealed specimens, the initial compressive strengths increased from approximately 29 MPa to 75 MPa (for GP 01 to GP 05) and from 23 MPa to 58 MPa (for GP 06 to GP 10). This could be due to the inclusion of more sodium silicate as also reported previously [18,41,66]. Silica gel, which favors the geopolymerization reaction, also has the capacity to accelerate this process by initiating the polymerization reaction of materials, resulting in a high early strength. It was also reported that [67] the use of sodium silicate improved the geopolymerization process by accelerating the dissolution of the fly ash. Furthermore, it was established that enhancing the levels of sodium silicate increased the $SiO_2$ to $Al_2O_3$ ratio, which in turn resulted in the increased number of Si–O–Si bonds, and this could lead to increased strengths [68].

However, it can be clearly noted that the initial compressive strength readings of the unsealed samples were much less consistent compared to that of the sealed samples. Unsealed GP 01, 02, and 03 specimens recorded lower strengths, and GP 04 and 05 exhibited higher strengths as the alkaline solution to the FA ratio increased from 0.4 to 0.57. The lower readings for GP 01, 02, and 03 specimens could be attributed to insufficient levels of fluids for the dissolution of solids, and for the formation of the gel structure, brought about by early dehydration processes. In contrast, the higher strength readings of GP 04 and 05 can be due to the formation of denser microstructures, with less pores, hence resulting in higher compressive strengths. This favorably compares with previous findings [69] where it was reported that higher strengths could be achieved at an alkaline solution to FA ratio of 0.4, compared to a ratio of 0.5–0.8.

Severe cracking was observed in unsealed Gladstone FA-based GP upon exposure to elevated temperatures (Figures 5 and 6). On the other hand, the results obtained for sealed Gladstone FA-based GP were far more promising compared to the unsealed specimens after the exposure to heat. For these specimens, no cracking was seen in the specimens GP 01–05 and only a mild degree of cracking in GP 06–10 after exposure 400 °C and 800 °C (Figures 7 and 8). This is in line with findings from another literature precedent [61], where it was also reported that initial evaporation severely hampers the evolution in the strength

of the specimens, thus were more likely to crack due to differential thermal gradients between the inside and outside of the samples.

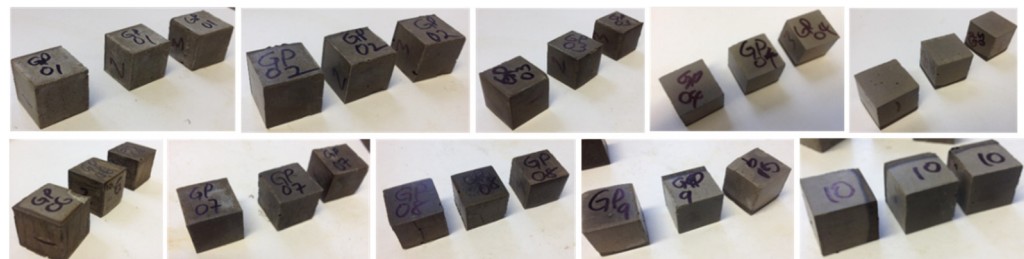

**Figure 5.** Pictures of unsealed Gladstone FA-based GP specimens—before exposure to an elevated temperature.

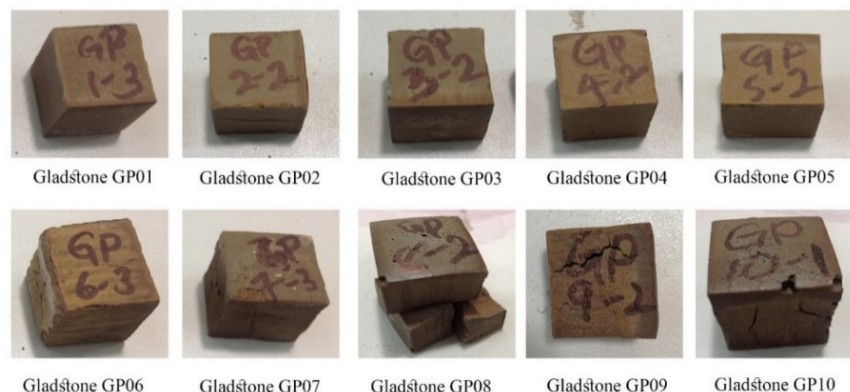

**Figure 6.** Pictures of unsealed Gladstone FA-based GP specimens—after exposure to 800 °C.

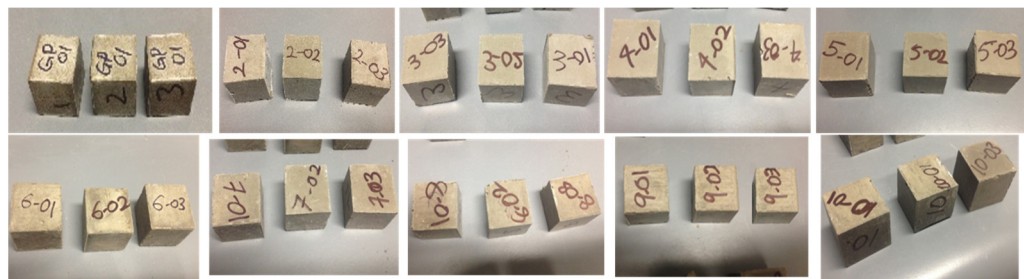

**Figure 7.** Pictures of sealed Gladstone FA GP specimens—before temperature exposure.

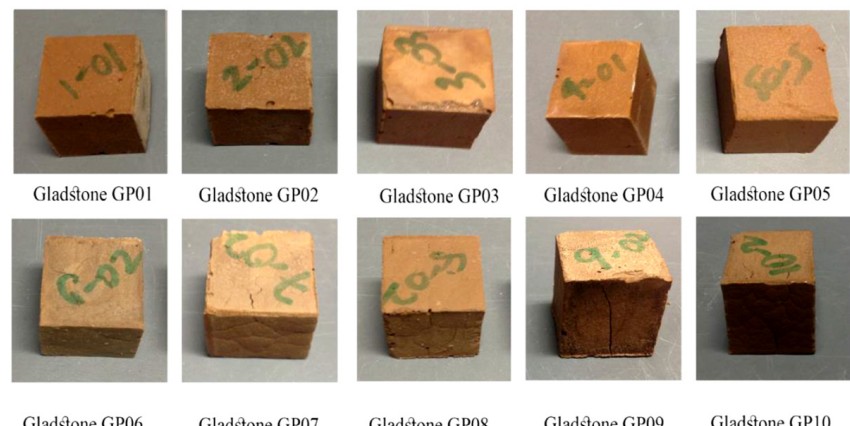

**Figure 8.** Pictures of sealed Gladstone FA-based GP specimens–after exposure to 800 °C.

As previously mentioned, this could also be attributed to the lower degrees of hydration of the matrices, which could lead to the breakdown of the granular structure of the matrices, thus restricting them to evolve into a more semicrystalline form. Subsequently, this effect could cause severe cracking when exposed to differential thermal gradients, especially at elevated temperature levels. Previous studies have also reported that an initial evaporation of fluids could hinder the continuous and uniform reorganization of polycondensation processes. This could in turn hinder further development in strength and structural integrity [61]. Furthermore, earlier dehydration processes can lead to less denser structures within the material, which can result in lower compressive strengths. Such effects were also reported previously [61,70], where it was also indicated that the inhibition of the ongoing geopolymerization process could result in less denser matrices, and ones with higher porous structures. These could lead to lower compressive strengths.

After exposure to 400 °C, unsealed Gladstone FA-based samples recorded an average compressive strength of approximately 28 MPa, while sealed samples recorded an average compressive strength of approximately 43 MPa. After the exposure to a temperature of 800 °C, unsealed Gladstone FA-based samples recorded an average compressive strength of approximately 18 MPa, while sealed samples recorded an average compressive strength of approximately 24 MPa. Tables 4 and 5 provide the data of the residual strength of unsealed and sealed samples, and Figures 9 and 10 show a graphical representation of these results.

**Table 4.** Values of average residual strengths (MPa): unsealed Gladstone FA-based GP cubes.

| Sample ID | 400 °C | STDEV | 800 °C | STDEV | Thermal Cracking 400 °C | Thermal Cracking 800 °C |
|---|---|---|---|---|---|---|
| GP01 | 15.14 | 6.75 | 24.27 | 4.79 | No | No |
| GP02 | 26.40 | 3.30 | 22.00 | 5.25 | No | Yes |
| GP03 | 26.19 | 3.48 | 22.29 | 6.04 | No | Yes |
| GP04 | 31.73 | 7.06 | 23.33 | 4.27 | No | Yes |
| GP05 | 52.55 | 3.12 | 25.20 | 4.06 | No | Yes |
| GP06 | 13.44 | 4.65 | 15.84 | 4.86 | Yes | Yes |
| GP07 | 24.48 | 5.36 | 12.11 | 3.04 | Yes | Yes |
| GP08 | 26.35 | 5.85 | 12.59 | 5.04 | Yes | Yes |
| GP09 | 20.81 | 5.44 | 12.37 | 3.52 | Yes | Yes |
| GP10 | 30.69 | 4.00 | 13.81 | 5.20 | Yes | Yes |
| Minimum | 13.44 | 3.12 | 12.11 | 3.04 | N/A | N/A |
| Maximum | 52.55 | 7.06 | 25.20 | 6.04 | N/A | N/A |
| Average | 27.81 | | 18.43 | | N/A | N/A |

**Table 5.** Values of average residual strengths (MPa) for sealed Gladstone FA-based GP cubes.

| Sample ID | 400 °C | STDEV | 800 °C | STDEV | Thermal Cracking 400 °C | Thermal Cracking 800 °C |
|---|---|---|---|---|---|---|
| GP01 | 35.62 | 1.99 | 26.22 | 4.76 | No | No |
| GP02 | 47.80 | 3.22 | 35.31 | 3.35 | No | No |
| GP03 | 54.42 | 0.73 | 48.05 | 1.23 | No | No |
| GP04 | 74.14 | 1.49 | 38.29 | 0.74 | No | No |
| GP05 | 56.91 | 3.81 | 36.49 | 5.07 | No | No |
| GP06 | 17.71 | 1.47 | 10.53 | 3.61 | Yes | Yes |
| GP07 | 26.46 | 3.72 | 7.69 | 2.96 | Yes | Yes |
| GP08 | 39.99 | 3.24 | 13.37 | 2.82 | Yes | Yes |
| GP09 | 29.33 | 2.52 | 11.47 | 3.47 | Yes | Yes |
| GP10 | 39.18 | 2.83 | 9.49 | 2.16 | Yes | Yes |
| Minimum | 17.71 | 0.73 | 7.69 | 0.74 | N/A | N/A |
| Maximum | 74.14 | 3.81 | 48.05 | 5.07 | N/A | N/A |
| Average | 42.78 | | 24.39 | | N/A | N/A |

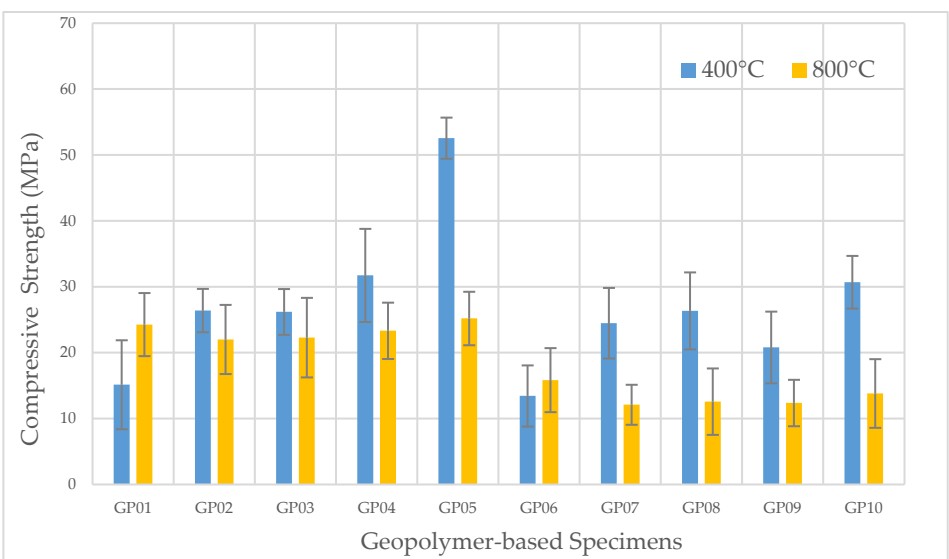

**Figure 9.** A plot of residual strength for unsealed Gladstone FA-based GP specimens.

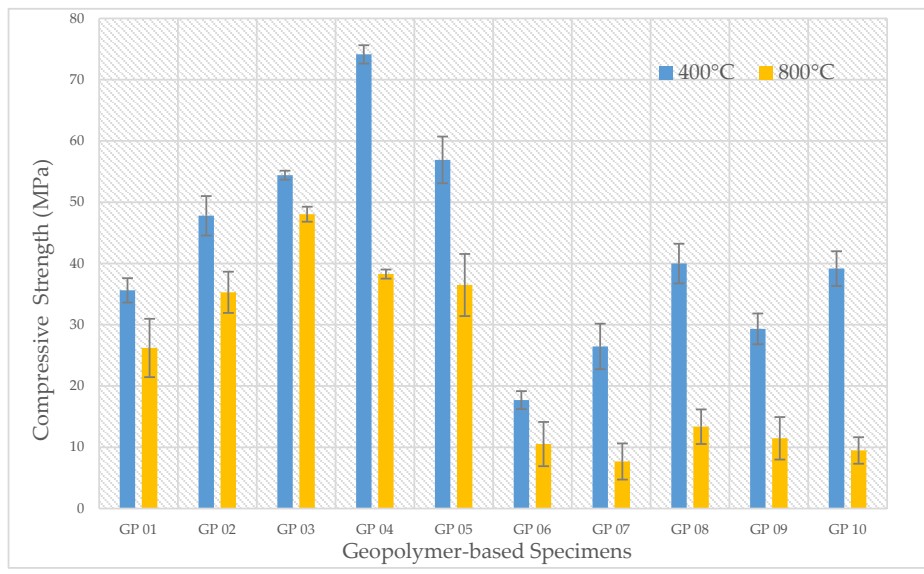

**Figure 10.** A plot of residual strength for sealed Gladstone FA-based GP specimens.

It was also noted that as the temperature increased to 400 °C, the majority of the Gladstone FA-based GP samples having a lower ratio of alkaline solution to FA (ca. 0.4), were shown to exhibit higher residual strengths. This increase could be due to less voids within the paste matrix, which would provide thermal stability, as reported previously [70–72]. Furthermore, unsealed GP 01 and 06 specimens recorded an increase in residual strength over a range of 400 °C to 800 °C, which could be attributed to the lowest $Na_2SiO_3/NaOH$ ratio (ca. 0.5) of these two mixtures (GP 01 and 06). As sodium silicate has a higher thermal resistance, correspondingly at a higher alkali activator ratio, strength gains can be expected, even at higher temperature levels.

It is to be noted here that the majority of the Gladstone FA specimens recorded higher initial compressive strengths compared to the Gladstone/Callide FA-based GP specimens. This could be attributed to the variances in particle sizes—Gladstone FA has a fineness of 86% passing the 45 μm sieve compared to 80% of the Gladstone/Callide FA. This essentially points to the fact that the former has finer particles compared to the latter. It is also reported that [73] finer particles have a higher rate of dissolution during the geopolymerization process, thus resulting in higher compressive strengths. The difference in the composition

of the aluminosilicates has also been reported to affect the strength retention after exposure to high temperature levels [74]. A higher conversion of FA to amorphous aluminosilicate gels, and hence, a better retention of strength, has been reported at a higher Si/Al ratio. This supports the findings of the present study, where higher strengths were recorded from Gladstone FA-based GPs, which had a Si/Al ratio of 2.0 compared to 1.8 of the Gladstone/Callide FA.

Table 6 and Figure 11 provide a tabulated and graphical comparison of the initial compressive strength between the two types of FA. In addition, GP 08 of the Gladstone/Callide FA-based specimens, which had a silicate to hydroxide ratio of 1.75 and a hydroxide solution to fly ash ratio of 0.57, resulted in the highest initial strength reading of approximately 61 MPa. This could be due to the excessive silicate (i.e., beyond a ratio of 1.75), which can inhibit the geopolymerization process through the precipitation of the Al–Si phase [75].

**Table 6.** Compressive strengths (MPa) at 24 h among sealed Gladstone and Gladstone/Callide FA GP samples *.

| Sample ID | Gladstone FA GP | STDEV | Gladstone/Callide FA GP | STDEV |
|---|---|---|---|---|
| GP01 | 28.76 | 4.67 | *— | *— |
| GP02 | 41.31 | 3.28 | *— | *— |
| GP03 | 67.06 | 3.31 | *— | *— |
| GP04 | 67.95 | 1.34 | 56.75 | 2.35 |
| GP05 | 74.48 | 3.41 | 49.91 | 1.17 |
| GP06 | 22.67 | 3.64 | 41.15 | 3.52 |
| GP07 | 41.98 | 2.57 | 43.35 | 0.60 |
| GP08 | 54.77 | 2.12 | 61.38 | 1.99 |
| GP09 | 55.38 | 1.32 | 54.53 | 1.74 |
| GP10 | 58.13 | 1.72 | 51.99 | 1.93 |
| Minimum | 22.67 | 1.32 | 41.15 | 0.60 |
| Maximum | 74.48 | 4.67 | 61.38 | 3.52 |
| Average | 51.25 | | 51.29 | |

* Due to poor workability conditions, the corresponding values for GP01, GP02, and GP03 could not be measured.

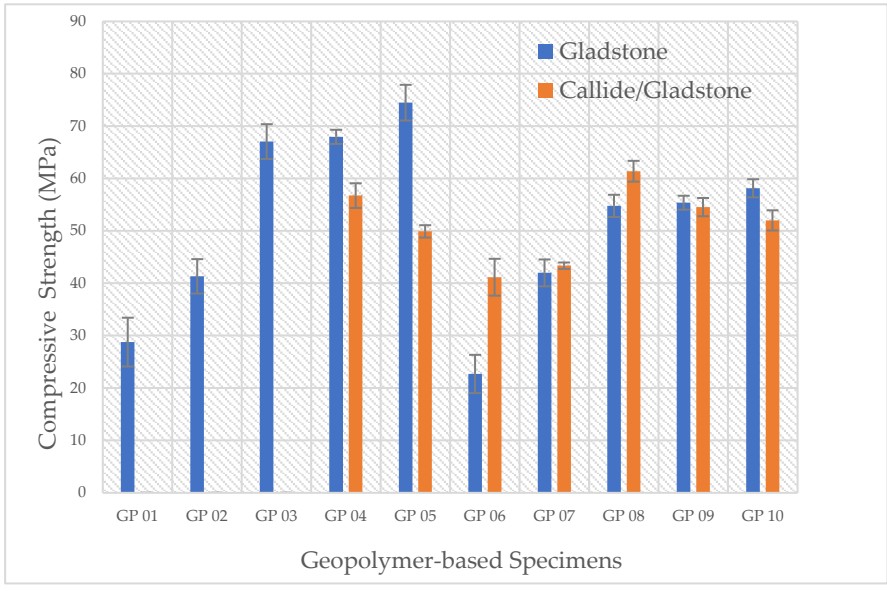

**Figure 11.** Initial compressive strength among the sealed Gladstone and Gladstone/Callide FA-based GP samples.

The initial results pertaining to the strength of the samples clearly indicated that the majority of the Gladstone FA-based GP pastes performed better amongst the two classes of materials. However, while the majority of the initial compressive strengths of the Gladstone/Callide samples were low, the residual strengths were considerably high, with a striking maximum strength of approximately 90 MPa (Table 7) compared to approximately 74 MPa for Gladstone FA-based specimens. Similar results were previously reported [72,76,77], where relatively low strength GP mixtures were observed to produce higher thermal performance and vice versa. It was also reported that this condition can be attributed to the chemical constitution, and microstructural changes occurring at higher temperatures. Samples consisting of Gladstone/Callide FA have a higher level of silicon and aluminum than Gladstone FA (Table 1). This could strongly influence the geopolymerization reactions, possibly forming the gel layer at the surface. Generally, a sintering of this gel phase is effected as temperature increases, and this in turn facilitates higher cohesive strengths, and could result in more homogeneous and denser microstructures [76,78].

**Table 7.** Values of average residual strengths: Gladstone/Callide FA-based GP cubes.

| Sample ID | 400 °C | STDEV | 800 °C | STDEV | Thermal Cracking 400 °C | Thermal Cracking 800 °C |
|---|---|---|---|---|---|---|
| GP 04 | 76.43 | 4.13 | 25.91 | 1.00 | No | Yes |
| GP 05 | 90.02 | 1.04 | 30.43 | 2.31 | No | Yes |
| GP 06 | 56.51 | 2.52 | 18.51 | 3.68 | No | Yes |
| GP 07 | 56.62 | 3.34 | 19.84 | 3.71 | No | Yes |
| GP 08 | 47.73 | 0.37 | 14.43 | 3.73 | No | Yes |
| GP 09 | 59.10 | 3.00 | 13.61 | 1.37 | No | Yes |
| GP 10 | 41.03 | 3.32 | 15.22 | 0.90 | No | Yes |
| Minimum | 41.03 | 0.37 | 13.61 | 0.90 | N/A | N/A |
| Maximum | 90.02 | 4.13 | 30.43 | 3.73 | N/A | N/A |
| Average | 62.05 | | 20.22 | | N/A | N/A |

In addition, a higher degree of thermal conductivity is achieved with enhanced contents of silicon and aluminum, resulting in lower thermal gradients between the inside and outside of the specimen. Therefore, the Gladstone/Callide FA-based material has the ability to produce higher thermal energy bearing capabilities. However, the reverse effect can be expected with a lower level of Si-Al minerals (i.e., poor bonding properties and a higher thermal incompatibility within the specimen emanating from a reduced conductivity). Similar deductions have been reported elsewhere [79,80]. These samples with low initial strengths were observed to display higher levels of ductility and improved strength. The reverse was found to be true in the case of samples that were relatively brittle. However, when considering the overall thermal performance of the GP specimens made with either Gladstone or Gladstone/Callide FA, it can be clearly seen that they exhibited relatively good thermal stability. Similar results were reported previously [81], where the failure of GPs under thermal conditions is in fact influenced by the strength and stress–strain properties.

Generally, mass loss can be identified as an important factor when evaluating the thermal performance of concretes. The rate of mass loss upon reaching 150–200 °C was high, reported to be due to the loss of physically bound water (free water content), after which the rate somewhat stabilized, with the decomposition of $CaCO_3$ occurring within a temperature range of 600–800 °C [47,79,80,82,83]. Table 8 presents the percentage mass loss readings after exposure to 400 °C and 800 °C of the unsealed and sealed Gladstone FA-based GP samples and sealed Gladstone/Callide FA-based GP samples.

**Table 8.** Average percentage of mass losses.

| Sample ID | Unsealed Gladstone FA-Based GP Cubes | | Sealed Gladstone FA-Based GP Cubes | | Sealed Gladstone/Callide FA-Based GP Cubes | |
|---|---|---|---|---|---|---|
| | 400 °C | 800 °C | 400 °C | 800 °C | 400 °C | 800 °C |
| GP 01 | 5.19 | 6.71 | 14.10 | 15.88 | - | - |
| GP 02 | 6.56 | 7.41 | 14.61 | 15.34 | - | - |
| GP 03 | 7.02 | 7.66 | 14.08 | 14.86 | - | - |
| GP 04 | 8.41 | 9.41 | 15.09 | 15.37 | 14.63 | 16.46 |
| GP 05 | 8.57 | 9.33 | 15.08 | 15.79 | 14.15 | 16.13 |
| GP 06 | 6.55 | 7.57 | 22.39 | 21.68 | 17.82 | 19.57 |
| GP 07 | 7.03 | 7.91 | 20.33 | 20.99 | 18.21 | 20.07 |
| GP 08 | 7.21 | 8.19 | 19.27 | 19.52 | 18.23 | 20.66 |
| GP 09 | 18.18 | 18.24 | 19.66 | 19.31 | 18.68 | 20.70 |
| GP10 | 18.13 | 18.50 | 19.75 | 20.04 | 17.27 | 17.90 |
| Minimum | 5.19 | 6.71 | 14.08 | 14.86 | 14.15 | 16.13 |
| Maximum | 18.18 | 18.50 | 22.39 | 21.68 | 18.68 | 20.70 |
| Average | 9.55 | 10.09 | 17.43 | 17.88 | 17.00 | 18.79 |

It can be noted that the majority of the unsealed Gladstone FA-based GP specimens exhibited average mass losses (up to 10%) when subjected to elevated temperatures (400 °C and 800 °C). Sealed Gladstone FA-based GP specimens, on the other hand, recorded consistent losses, with an average mass loss of approximately 17% after being exposed to similar temperatures. The low mass loss percentage in the case of unsealed samples can be attributed to the presence of relatively lower proportions of water content in the unsealed samples. It can also be noted that large losses were recorded from the latter half of the samples (i.e., for GP 06–10). They had a higher alkaline solution to FA ratio (ca. 0.57) compared to the former half (i.e., for GP 01–05). As previously mentioned, above 1.75 (ratio of $Na_2SiO_3$ to NaOH), the silicate in the mixture inhibited the geopolymerization reactions [75]. Therefore, the two pertinent ratios (i.e., alkaline solution to FA ratio and $Na_2SiO_3$ to NaOH), in combination, can lead to higher mass losses. It was also reported [84] that average mass losses of FA-based GP pastes was about 19 and 20% at temperatures of 600 °C and 800 °C, respectively, which are similar to the findings of the present study.

The values of mass losses for the Gladstone/Callide FA-based GP specimens were approximately 17% and 19% after being exposed to elevated temperatures of 400 °C and 800 °C, respectively. The corresponding losses recorded for the Gladstone FA were somewhat similar. It can be assumed here that changes in the chemical constitution is not a key governing factor responsible for the loss of mass after heat exposure. In addition, GPs have been reported to remain chemically stable up to temperatures of about 800 °C [32].

## 4. Conclusions

This study focused on investigating the effects of initial surface evaporation on the performance of FA-based geopolymer pastes. Evaporation was controlled optionally for some test specimens by placing them in a sealed cover during the curing process. In addition, the thermal performance of two different types of Class F (low calcium) fly ash (namely Gladstone and Gladstone/Callide) was also investigated.

The main points emanating from the preset investigation can be depicted as follows:

- It can be seen that the degree of initial surface evaporation has a major effect on the final performance of the GP specimens, where sealing of the samples during the curing process, and hence limiting excess initial surface evaporation, produced more consistent results as well as much higher strength readings.

- Unsealed specimens exhibited an approximately 25% lesser value for the average initial strength, and approximately 35% and 25% lesser readings for the average residual strength readings at 400 °C and 800 °C, respectively. In addition, in the case of unsealed specimens, a comparably higher degree of thermal cracking and splitting were observed compared to the sealed specimens.
- The initial strength readings also increased from approximately 15–58 MPa (GP 01–05) and 40–55 MPa (GP 06–10) for unsealed samples and 29–74 MPa (GP 01–05) and 22–58 MPa (GP 06–10) for sealed samples as the $Na_2SiO_3$/NaOH ratio increased from 0.5 to 2.5, presumably due to the increased levels of sodium silicate. Furthermore, an enhanced $SiO_2$ to $Al_2O_3$ ratio is likely to cause an acceleration of the geopolymerization process, and hence resulted in relatively higher early strengths for the samples.
- Majority of both Gladstone and Gladstone/Callide FA GP specimens exhibited higher initial strength readings at relatively lower alkaline solution to FA ratios. A maximum reading of approximately 75 MPa for the Gladstone and 57 MPa for the Gladstone/Callide FA GP specimens were recorded at 0.4 alkaline solution to FA ratio. This can be attributed to their comparatively denser microstructures, thus forming more homogenous materials with fewer number of pores.
- In the case of the Gladstone/Callide FA specimens, large increments in strength, with a striking maximum of approximately 90 MPa, was observed compared to those made from Gladstone FA after being exposed to 400 °C. These variations can be attributed to a relatively higher level of silicon and aluminum, which in turn resulted in enhanced internal strength, primarily emanating from higher degrees of sintering, especially, at elevated temperatures. A higher degree of thermal conductivity can be thought to arise due to increased levels of silicon and aluminum in these specimens, which also will aid in reducing the differential thermal gradient. However, noticeable losses in strength were recorded for several of the Gladstone/Callide FA-based samples after 400 °C, and this also resulted in severe cracking compared to the Gladstone FA-based samples. This may be attributed to the occurrence of an increased level of pore pressure within the samples, causing higher stresses at elevated temperatures. Hence, it can be stated that Gladstone FA-based geopolymers exhibited excellent thermal stability compared to Gladstone/Callide FA-based geopolymers.
- The Gladstone/Callide FA-based mixtures exhibited relatively quicker initial setting times (ca. 2–4 min) compared to the Gladstone FA-based mixtures, which remained in liquid form for over 30 min.

**Author Contributions:** Conceptualization, M.G., S.F. and P.J.; Methodology, M.G. and T.K.; Validation, M.G., S.F., P.J. and T.K.; Formal analysis, T.K.; Investigation, M.G., S.F., P.J. and T.K.; Resources, M.G.; Data curation, M.G., S.F., P.J. and T.K.; Writing—original draft preparation, M.G., P.J. and T.K.; Writing—review and editing, P.J.; Supervision, M.G., S.F. and P.J.; Project administration, M.G., S.F. and P.J.; Funding acquisition, M.G. All authors have read and agreed to the published version of the manuscript.

**Funding:** This research received no external funding.

**Institutional Review Board Statement:** Not applicable.

**Informed Consent Statement:** Not applicable.

**Data Availability Statement:** Data are contained within the current article.

**Acknowledgments:** The authors are grateful for the technical support provided by Lyndon Macindoe and Philip Dunn, Institute for Sustainable Industries and Liveable Cities.

**Conflicts of Interest:** The authors declare no conflict of interest.

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
