# Peer review of "Effects of Initial Surface Evaporation on the Performance of Fly Ash-Based Geopolymer Paste at Elevated Temperatures"

_applsci, doi:10.3390/app12010364_

Round 1

Reviewer 1 Report

Dear respected editor,

The paper presents the interesting experimental research work on geopolymer technology, but, it contains a range of issues that detract from its overall quality as shown below:

  1. The quality of the English in the paper needs to be improved.
  2. There is no clear distinguish between geopolymer paste and geopolymer concrete in this paper. Since no aggregates have been used to produce the geopolymer in this experimental study, the final mixture would be geopolymer binder (or paste) and not geopolymer concrete! Therefore, anywhere in the manuscript that geopolymer concrete have been wrongly used such as in the title, line 107 and etc. must be modified!
  3. Table 2 should be double-checked! For instance, row 2 should be GP-0.40; 1.00 instead of GP-0.4; 1.00 and also the last row is repetitive and must be eliminated!
  4. Line 153, the compressive and residual strengths are categorized as mechanical properties and not physical properties! Please modify!
  5. Line 154, it says: “Unsealed samples were placed in the oven at 60 C immediately after curing”! But seems it should be: “Unsealed samples were placed in the oven at 60 C immediately after casting”. Please double check!
  6. Line 169, Why setting times have been qualitatively (by visual observation) and not quantitatively (for example based on ASTM) measured!
  7. Line 227, it says: “the highest value was for GP 10”, which is in contradiction with the results presented in Table 3 and Figure 4! Please explain!
  8. There is some repetitive information that need to be eliminated; For example, the data presented in Table 3 is the same as that in Figure 4!
  9. The authours need to explain further about the exceptions in the trends of results; such as: 

Figure 4, regarding the unsealed samples, when the Alkaline to FA ratio increases from 0.4 to 0.57, in some sodium silicate to sodium hydroxide ratios, the compressive strength increases and in some others decreases! Please expalin!

Figure 9, the compressive strength of GP01 (in both Alkaline to FA ratios of 0.4 and 0.57) at 800 C curing is higher than that for 400 C curing! Please elaoborate!

Author Response

Please see enclosed.

Reviewer 2 Report

The authors present an interesting study regarding the Effects of Initial Surface Evaporation on the Performance of Fly Ash-based Geopolymer Concrete at Elevated Temperatures. The results can be very useful for other researchers and English usage is generally correct and comprehensible.

The paper is well organized. The introduction describes the state-of-the-art with respect to the previous and present background for this area.

The research design and methods are clearly stated. The results are well presented and explained, based on their own experiments, and the conclusions thoroughly supported by the results presented in the article or referenced in the secondary literature. Tables and figures are very clear and understandable.

Based on the above appreciations, I want to congratulate the authors for their work and I consider that this article could be published in the present version.

Author Response

Please see enclosed.

Reviewer 3 Report

The article entitled: "Effects of Initial Surface Evaporation on the Performance of Fly Ash-based Geopolymer Concrete at Elevated Temperatures" is in line with Applied Science journal. The article based on original research. The topic is up-to-date. The article is overall well planned, however, the manuscript requires major changes, such as:

  • Introduction: Please clarify the text in lines 42-49 (repetitions), what about other sources? (for example: https://www.scopus.com/record/display.uri?eid=2-s2.0-85108688359&origin=resultslist).
  • Introduction: line 50, ref [20] – please add information about mentioned temperatures.
  • Introduction: Lack of detailed analysis of literature for investigation provided for elevated temperatures, for example: https://link.springer.com/article/10.1186%2Fs40069-018-0267-2 or https://www.mdpi.com/1996-1944/13/23/5548 and many others.
  • Introduction: add to the last paraph information about novelty aspects of provided research.
  • Part 2.1.: please clarify the proportion used two kinds of FAs.
  • Table 2. What is the difference between first and last line in composition?
  • Part 2.4.: the information about samples dimensions is needed.
  • Table 3 and others with mechanical properties: please add information how many samples were tested in each serie. Statistical information (error; standard deviation) are required).
  • Figure 4: add bar errors.
  • Figure 9 and 10: add bar errors.
  • Results and discussion: there is a lack of DISCUSSION with the state-of-the-art, including scientific literature from last few years.
  • Conclusion part should be more detailed, the obtained measurable results should be presented and application aspect should be commented.

Author Response

Please see enclosed.

Round 2

Reviewer 3 Report

The article entitled: "Effects of Initial Surface Evaporation on the Performance of Fly Ash-based Geopolymer Concrete at Elevated Temperatures" was significantly improved. It reequires only slight change - stress the novelty aspects of provided research (Introduction part).

Author Response

See enclosed
